# Attached but Lonely: Emotional Intelligence as a Mediator and Moderator between Attachment Styles and Loneliness

**DOI:** 10.3390/ijerph192214831

**Published:** 2022-11-11

**Authors:** Dominik Borawski, Martyna Sojda, Karolina Rychlewska, Tomasz Wajs

**Affiliations:** Department of Psychology, Jan Kochanowski University, 25-029 Kielce, Poland

**Keywords:** loneliness, attachment styles, attachment anxiety, attachment avoidance, emotional intelligence, self-worth

## Abstract

(1) Background: The aim of the presented research was to examine if emotional intelligence (EI) could be regarded as a mechanism mediating the relationship between attachment and loneliness. The authors also tested the moderating role of this variable, investigating whether EI was a protective factor against loneliness in insecurely attached individuals. (2) Methods: In two cross-sectional studies (*N* = 246 in Study 1 and *N* = 186 in Study 2), participants completed a set of questionnaires measuring attachment styles, trait emotional intelligence, and loneliness. (3) Results: Both studies revealed a consistent pattern of results, indicating a dual role of EI in the relationships between attachment styles and loneliness. Firstly, EI was a mediator between attachment and loneliness—both anxious and avoidant attachment were associated with a low level of EI, which in turn translated into increased loneliness. Secondly, EI moderated the relationship between anxious attachment and loneliness. It turned out that the strength of the positive relationship between anxious attachment and loneliness decreased with an increase in EI. (4) Conclusions: These results point to EI as an underlying mechanism between insecure attachment and loneliness. They also suggest that EI is an important psychological resource protecting anxiously attached individuals against a subjective sense of social isolation.

## 1. Introduction

Loneliness is defined as subjectively perceived social isolation stemming from the frustrated need to belong and a sense of discrepancy between the expected and actually experienced quality of social contacts [1]. Thus understood, it is a risk factor for numerous mental health issues including depression [2], suicidality [3], anxiety [4], and psychological distress [5]. Loneliness has little in common with objective indicators of isolation, such as the number of friends or acquaintances [6]. Its subjective character induces researchers to look for psychological factors that lead people to perceive and evaluate their interpersonal relationships differently in similar circumstances [7,8]. 

One of the conceptualizations that provide a framework for understanding the source and development of both satisfaction with social relationships and a sense of social isolation is attachment theory [9]. Viewed through the lens of this theory, loneliness is a consequence of early childhood experiences associated with the non-satisfaction of the need for closeness and tenderness or with the unavailability of attachment figures [10]. Supporting this theory, research shows that both attachment anxiety and attachment avoidance are positively correlated with loneliness [9]. 

Although the attachment–loneliness link is already well documented, there are still few studies showing the underlying mechanisms and moderators of this relationship [11]. Based on both theoretical premises and the results of previous research, the studies presented in this article were an attempt to establish if emotional intelligence (EI) could be regarded as a mechanism mediating the relationship between attachment and loneliness. Additionally, we investigated if this variable was a potentially protective factor against a sense of loneliness in insecurely attached people. 

### 1.1. Attachment and Loneliness

According to Bowlby, each person has an inborn psychobiological system responsible for seeking support and closeness from caregivers [12]. The primary aim of this system is to provide the individual with a sense of security, and the importance of its role grows in conditions of objective or subjective threats. In such circumstances, the individual is motivated to increase closeness with significant others; in the early stages of development, this happens through the regulation of physical contact with caregivers, whereas in the subsequent stages, it takes place via internal representations of attachment figures. Although from the child’s perspective the importance of physical closeness with attachment figures decreases with age in favor of subjective belief about their availability, the character of symbolic representations of self and others develops on the basis of earlier tangible interactions with actual others [13]. The quality of the primary bond with caregivers, formed in early childhood, especially the degree to which the attachment figures were available, responsive, and supportive, is reflected in internal representations of self and others. Thus, these original relationships with significant others become prototypes for subsequent interpersonal relationships. Interactions saturated with warmth and intimacy translate into a sense of security, a sense of self-worthiness, and positive expectations regarding their partners in close relationships in adult life. In contrast, when significant others are absent or insensitive to the child’s needs, the child develops attachment styles referred to as insecure, different from those based on seeking closeness [9]. Two of these are usually mentioned—namely, attachment-related anxiety and avoidance [14]. The former, anxiety, develops in the course of the child’s experience of uncertainty about the availability of the attachment figure. It manifests itself in an increased need for closeness and intimacy, accompanied by an intense fear of rejection rooted in low self-esteem. The other style, avoidance, develops when the child experiences unavailability of the attachment figure in situations of danger or experiences caregivers’ insensitivity to his or her needs. What is characteristic of this attachment style is, above all, a negative view of human nature, translating itself into a discomfort that accompanies dependence on others, lack of trust, and emotional distance from interaction with partners. Its essence lies in the deactivation of the attachment system in such a way as to minimize the risk associated with other people’s unavailability in situations of danger. Although both styles can initially be functional, for example as reactions to unsatisfied interpersonal needs in a specific relationship (e.g., with the caregiver), the problem is that they easily become entrenched and are easily carried over into other relationships, where more adaptive attitudes would be based on openness and on seeking support and intimacy. Therefore, early experiences saturated with both insecure attachment patterns not only make it impossible to perceive closeness, depth, and intimacy in adult relationships with others but, on the contrary, may actually lead to the frustration of the sense of belonging and to a subjective feeling of separation from others—namely, to loneliness [9]. In the case of anxiously attached individuals, loneliness may stem from an excessive desire to compensate for low self-esteem by gaining other people’s acceptance combined with a high level of rejection anxiety. In the case of avoidantly attached individuals, the key mechanism seems to be a negative view of human nature and a lack of interpersonal trust—a mindset that constitutes an important predictor of loneliness [9,15].

### 1.2. In Search of the Mechanisms Underlying the Attachment–Loneliness Link

The internal representations of self and others, mentioned above, function as the key underlying mechanism of the attachment–loneliness link. From this perspective, it is assumed that early childhood experiences with caregivers shape the internal models of self and others, which in turn become prototypes of subsequent interpersonal relationships and the experiences resulting from them (such as loneliness) [16]. A different, competing explanation of the link between attachment and loneliness is the level of social and emotional skills lowered due to insecure attachment, translating into a subjective sense of social isolation. In this context, the results of previous studies suggest that the relationship between insecure attachment and loneliness is mediated by low social skills [11] and by a weakened tendency to “catch” other people’s positive emotions [17] (although the latter is applied only to avoidantly attached participants). A different study, conducted by Marks et al. [18], suggests that another mediating mechanism may be emotional abilities understood as emotional intelligence (EI). These authors found that EI mediated the relationships between insecure attachment and subjective mental health outcomes, whose indicators included social dysfunction. At this point, EI is conceptualized as an ability or as a trait [19]. In the former case, it refers to abilities associated with recognizing, expressing, understanding, and regulating emotions, measured using objective tests of maximum performance. The latter concept, trait EI (as investigated in the already cited study by Marks et al.), is understood as self-perceptions of emotional abilities, that is, “how good we believe we are in terms of understanding, regulating, and expressing emotions in order to adapt to our environment and maintain well-being”, [19] (p. 335), which Petrides calls emotional self-efficacy; thus defined, it is measured using self-report questionnaires. Trait EI is intended to capture the degree to which abilities manifest themselves in the individual’s everyday experiences, including those in relationships with others [20]. For this reason, it was this concept that we relied on when operationalizing EI in both of our studies. However, regardless of approach (trait or ability) and regardless of the results reported by Marks et al. [18], viewing EI as a potential mediator between attachment and loneliness is also theoretically justified. It is assumed that the attachment styles forming in the early stages of development may influence the later development of EI (both as a trait and as an ability). Thus, secure attachment patterns resulting from the child’s experience of the availability of attachment figures, their sensitivity to the child’s needs, and emotional syntony with the caregiver favor the development of emotional self-awareness and adaptive emotional self-regulation skills [21]. Early insecure attachment experiences, by contrast, impair the proper recognition, understanding, and regulation of emotions [22]. In anxiously attached individuals, low emotional self-efficacy is the outcome of earlier interpersonal experiences, particularly the sense of uncertainty stemming from the fact that their attempts to obtain comfort and support from their caregivers brought the desired results on some occasions but not on others. As a result of this lack of control over the environment, they developed hypervigilance to threat cues, which took the form of maximizing both the very experience and the expression of negative emotions [23]. In practice, this manifests itself in a focus on negative information and events, which in turn triggers negative thoughts and ruminations, giving a secondary boost to anxiety [24]. According to Mikulincer and colleagues, however, the increased expression of negative emotions is supposed to attract caregivers’ attention and induce them to soothe these emotions [23,24]. In the case of avoidant individuals, problems with adaptive emotional regulation stem from an increased tendency to deactivate the attachment system associated with the unavailable or rejecting attachment figure. In practice, this manifests itself in striving to minimize experiences of negative affect and in turning attention away from threat cues [23]. Avoidantly attached people do not want to depend on others, which is why they suppress or repress all negative emotions that remind them of their own vulnerability. On the other hand, they have lowered abilities to empathize with other people and take their perspective. Thus, theoretically, problems with EI in the case of anxiously attached people consist in lowered ability to control negative affect, while in avoidantly attached people, they mainly take the form of difficulties in recognizing and understanding other people’s emotions. Importantly, both insecure attachment styles interfere with individuals deriving emotion regulation benefits from close interpersonal relationships, which may expose them to a sense of loneliness. Accordingly, previous research shows that EI prospectively predicts the level of loneliness [25,26].

### 1.3. Moderating Role of EI

Potentially, EI may be not only a mediator but also a moderator of the relationship between attachment and loneliness. More specifically, assuming—with Petrides et al. [19]—that EI defined as emotional self-efficacy can be developed during the life span, one can expect that its high level may be a factor protecting individuals with both insecure attachment styles from the experience of a subjective sense of social isolation in adult life. This would mean that even unfavorable early attachment-related experience does not doom individuals to loneliness if, in further stages of development, they manage to acquire skills associated with the appropriate comprehension, expression, and management of their emotions. There are data that justify this kind of prediction. First, previous studies showed that EI buffered various kinds of stress [27], and loneliness is, undeniably, a kind of social distress. Moreover, a recent investigation by Quintana-Orts et al. revealed that EI was a buffer against loneliness in adolescents experiencing cyber-victimization [28]. In the case of anxiously attached individuals, EI may enable more effective regulation of rejection anxiety and thus make it possible to establish more interpersonal contacts. On the other hand, it may facilitate recognizing other people’s emotions, thereby facilitating the selection of potential interaction partners and preventing rejection. In the case of avoidantly attached people, EI may help reduce the excessive tendency to rely on oneself only; it may also correct the previously acquired negative models of others and improve the abilities associated with taking other people’s perspectives. In this way, it may potentially open up new possibilities for building social connections with others for them, preventing loneliness.

### 1.4. The Present Research

The aim of the research presented in this article was to examine both the mediating role and the moderating role of EI in the relationship between attachment styles and loneliness. 

Based on the premises presented above, we formulated the following hypotheses:

**H1.** 
*Both insecure attachment styles (anxious and avoidant attachment) will be positively related to loneliness.*


**H2.** 
*Emotional intelligence will mediate the relationships between both anxious and avoidant attachment styles and loneliness; in other words, both anxious and avoidant attachment styles will be associated with decreased emotional intelligence, which in turn will translate into an increase in loneliness.*


**H3.** 
*The relationships between both attachment styles and loneliness will be moderated by the level of EI; namely, with an increase in the level of EI, the strength of the relationships between both anxious and avoidant attachment styles and loneliness will decrease.*


To test our hypotheses, we conducted two online cross-sectional studies. Both studies included the same measures of loneliness and EI. What they differed in was the use of different versions of the questionnaire assessing attachment styles and the fact that in the second study, replicating the mediation effect of EI, we included the already mentioned construct, internal models of self and others, while additionally controlling for the mediating role of self-worth and beliefs about the benevolence of the people.

## 2. Materials and Methods

### 2.1. Participants and Procedure

In both studies (*N* = 246 in Study 1 and *N* = 186 in Study 2), participants (aged 18–61 years, *M* = 22.64, *SD* = 6.78, and 18–52 years, *M* = 23.81, *SD* = 5.04, respectively) were recruited in Poland using snowball sampling. The study was conducted online via the Google Forms platform. The link to the survey was distributed via social media. Participation was voluntary—the subjects were informed that they could withdraw from the study at any time; they did not receive any remuneration. After providing informed consent, they were given access (i.e., a link) to the survey. All the required ethical standards were maintained. The study protocol was approved by the Research Ethics Board at the authors’ institution. The respondents were asked to provide demographic information: gender, age, relationship status, place of residence, education (only in Study 2), and employment status (only in Study 1). All participants were Polish residents. The sociodemographic characteristics of the samples are presented in Table 1. 

### 2.2. Measures

#### 2.2.1. Loneliness

In both studies, we assessed loneliness using the UCLA Loneliness Scale—Revised [29] as adapted into Polish by Kwiatkowska et al. [30]. This measure consists of 20 items (e.g., “I feel left out”), which participants rate on a 4-point Likert scale (1 = *never* to 4 = *often*). The items (half of them reverse-scored, e.g., “There are people I feel close to”) refer to individuals’ interpersonal experiences and enable the assessment of subjectively perceived social isolation. The reliability coefficients for the total loneliness score were 0.90 in Study 1 and 0.91 in Study 2. 

#### 2.2.2. Attachment Styles

In Study 1, to measure attachment styles, we used the Experiences in Close Relationships—Relationship Structures (ECR-RS) Questionnaire [31], as adapted into Polish by Marszał [32]. It is a self-report measure enabling the assessment of both anxious and avoidant attachment experiences in various types of interpersonal relationships. The questionnaire consists of nine items, each of them related to four targets: mother, father (or mother-like figure and father-like figure, respectively), romantic partner, and best friend (e.g., “I don’t feel comfortable opening up to this person” for attachment-related avoidance and “I’m afraid that this person may abandon me” for attachment-related anxiety). Subjects respond using a 7-point scale (1 = *strongly disagree* to 7 = *strongly agree*). For the purposes of our research, we used global attachment scores, that is, global avoidance and global anxiety ratings, being the mean scores computed across the four targets. Due to an error in the construction of the online survey form, Item 6 concerning avoidant attachment to the father was not included in the study. The Cronbach’s alpha coefficients for the Anxiety and Avoidance subscales were 0.86 and 0.89, respectively. In Study 2, we administered a different version of the ECR scale, assessing the same attachment styles, but this time limited exclusively to parental attachment. It was a version of the measure developed by Brennan et al., modified by Marchwicki [33]. While the original version assessed adult romantic attachment, Marchwicki’s is designed for the retrospective assessment of the styles of attachment to the mother and the father in childhood. The content of the items concerns the respondent’s relations with the parents when he or she was 11–12 years old. In each item, the original expressions such as “romantic partner” or “my partner” were replaced with “mother and father,” and the verb tense, originally present, was changed to past in the Polish translation. As in the case of ECR-RS, in Marchwicki’s measure items are rated on a 7-point Likert scale (1 = *strongly disagree* to 7 = *strongly agree*). The questionnaire consists of 40 questions, 19 of them referring to relations with the mother and 21 referring to relations with the father. As in Study 1, also in the case of this version of the measure, we used global scores on anxiety and avoidance attachment styles, calculated by averaging the scores on relationships with mother and father. The reliability coefficients for both subscales were acceptable: 0.87 for Anxiety and 0.93 for Avoidance. 

#### 2.2.3. EI

In both studies, we measured EI using the Trait Emotional Intelligence Questionnaire—Short Form (TEIQue-SF) [34], as adapted into Polish by Szczygieł et al. [35]. The questionnaire is a self-report measure and consists of 30 items (e.g., “Many times, I can’t figure out what emotion I’m feeling”), which subjects respond to on a 7-point scale (1 = *completely disagree* to 7 = *completely agree*). The Polish version of the questionnaire has a one-factor structure and measures the global level of trait emotional intelligence. The higher the score, the higher the level of trait EI. The reliability coefficients for global trait EI were acceptable in both studies, their values being 0.92 in Study 1 and 0.89 in Study 2. 

#### 2.2.4. Internal Representations of Self and Others

Additionally, Study 2 included the measurement of internal working models of self and others, operationalized as faith in and the strength of conviction about both self-worth and the benevolence of the people. To assess them, we used two subscales of the World Assumptions Scale (WAS) [36] as adapted into Polish by Załuski and Gajdosz [37], namely Self-Worth and Benevolence of the People. Each of them consists of 4 items. In the case of Self-Worth, the items refer to the belief that one is good, competent, and moral by nature; in the case of Benevolence of the People, they refer to the faith that other people are—by nature—good, gentle, helpful, protective, and trustworthy. In both subscales, participants rated the items on a 6-point Likert scale (1 = *strongly disagree* to 6 = *strongly agree*). In the case of the Self-Worth subscale, reliability was fully acceptable (α = 0.85), while in the case of Benevolence of the People, it was lower, although still nearly acceptable (α = 0.69).

### 2.3. Data Analysis

The data analysis procedure was similar in both studies. The first stage consisted in screening for missing data. In the first study, there were no missing answers. In the second study, because we were interested in global scores on attachment styles, we considered only those responses that concerned both parents. When the respondent was not able to complete a questionnaire concerning one of their parents, his or her responses were excluded from the analysis. This was the case with three participants. We also screened the data for potential multivariate outliers by checking the standardized residuals from the regression analyses reported below. Before commencing our research, we decided that observations would be excluded if their residual was more than three standard deviations from the predicted value; this was never the case in Study 1, and in Study 2, it was the case for one respondent. As a result, further analyses included data from 246 and 186 subjects in Studies 1 and 2, respectively. 

Next, we calculated descriptive statistics and correlations between the variables using JAMOVI software. To test Hypothesis 2, we performed mediation analyses (It should be noted that TEIQue-SF includes two items (i.e., #9: I feel that I have a number of good qualities, and #24: I believe I’m full of personal strengths) concerning self-esteem, a variable conceptually similar to self-worth. Therefore, to control for content overlap between TEIQue-SF and the Self-Worth subscale of the WAS, in Study 2, we performed alternative mediation analyses, with these two TEIQue-SF items excluded. Moreover, all the mediation and moderation analyses presented here were also performed in an alternative manner, with participants’ gender and age controlled for. All of these additional analyses yielded results essentially the same as those reported in the text. We present them in Appendix A) using jAMM add-on for JAMOVI, which allows for regression-based path modeling with observed variables (To obtain the observed variables in the tested models, we averaged the original ordinal variables (i.e., the responses scored on Likert-type ordinal scales) and treated them as measured on interval scales.). Because of this tool, we were able to apply advanced models controlling for both attachment styles as independent variables (Study 1 and Study 2, see Figure 1 and Figure 2), with loneliness as a dependent variable, and with multiple parallel mediators between them (i.e., EI, self-worth, and benevolence of the people at the same time; Study 2, see Figure 2). All the coefficients in the mediation analyses were estimated using the maximum likelihood method implemented in jAMM. Standard errors were based on the expected information matrix. Betas (β) were obtained as completely standardized parameters of the path model. 

In both studies, we also tested the moderating effect of IE using multiple regression analyses. We performed two such analyses in each study. In the tested models, a particular attachment style, EI, and the interaction term of these two variables were included as predictors of loneliness, while the other attachment style (i.e., the one not included in the interaction) was entered as a covariate (see Figure 3A,B). Both mediating and moderating effects were examined using the bootstrapping method that relies on 95% bias-corrected confidence estimates (10,000 bootstrapped resamples). In the case of significant moderating effects, in order to obtain a more nuanced picture of results, we performed simple slope tests and provided Johnson–Neyman regions for interaction effects using the interActive data visualization tool [38].

## 3. Results

Descriptive statistics for the variables included in both studies are presented in Table 2. The distribution of all variables in both studies was close to normal, as the absolute values of skewness and kurtosis did not exceed 1. Both in Study 1 (see Table 3) and in Study 2 (see Table 4), anxious and avoidant attachment styles were correlated negatively with EI and positively with loneliness. Loneliness and EI were negatively correlated with each other. Moreover, in Study 2, self-worth was negatively correlated with both attachment styles and loneliness and positively with EI. Benevolence of the people was negatively correlated with avoidant attachment and loneliness and positively associated with EI, but it was not significantly related to anxious attachment.

Both in Study 1 and in Study 2, the indirect effects involving EI as a mediator between both anxious and avoidant attachment and loneliness were significant (β = 0.21,95% CI [0.05, 0.12] for anxiety and β = 0.21, 95% CI [0.08, 0.18] for avoidance in Study 1, and β = 0.07, 95% CI [0.01, 0.06] for anxiety and β = 0.20, 95% CI [0.05, 0.11] for avoidance in Study 2, respectively; see Table 5 for details). In Study 2, in which we tested six mediation effects, the significant ones also included the two indirect effects involving self-worth, indicating a mediating role of this variable between both attachment styles and loneliness (β = 0.07, 95% CI [0.01, 0.06] for anxiety and β = 0.09, 95% CI [0.01, 0.06] for avoidance, respectively). Effects involving benevolence of the people as a mediator turned out to be nonsignificant (see Table 6). 

Moderation analyses revealed a similar pattern of results in both studies. While interaction effects for anxious attachment and EI were significant, interaction effects involving avoidant attachment did not predict the level of loneliness (see Table 7 and Table 8). This means that EI proved to be a moderator of the relationships between anxious attachment and loneliness; in both studies, the effect of anxiety attachment on loneliness was qualified by the level of EI. To obtain a more nuanced picture of moderation effects, we performed simple slope tests and provided Johnson–Neyman regions for the interaction effects found. In both studies, simple slope tests showed that the positive effect of anxious attachment on loneliness was the strongest at a low level of EI and gradually weakened with an increase in the level of this variable (see panel A in Figure 4 and Figure 5). Further exploration using the Johnson–Neyman technique (see panel B in Figure 4 and Figure 5) revealed that in Study 1, the effect of anxious attachment ceased to be significant even in participants who scored −0.2 SD above the mean or higher on EI (54.47%), while in the case of Study 2, it was no longer significant in those who scored 0.55 SD above the mean or higher on EI (30.65%). 

## 4. Discussion

The aim of our research was to examine the potential mediating and moderating roles of EI between attachment styles and loneliness. The results of the two cross-sectional studies revealed that both anxious and avoidant attachment styles were positive predictors of loneliness. These results are in line with previous investigations [9,17]. In the case of anxious attachment, the link with loneliness can easily be explained by the fact that early childhood experiences of anxiously attached people combine a strong need for acceptance from others with a fear of being rejected or abandoned. This combination probably favors the more frequent occurrence of the non-satisfaction of the need to belong, which is characteristic of loneliness. The link between avoidant attachment and loneliness is not so obvious, since avoidantly attached individuals are, theoretically, less dependent on others, and social distancing is not necessarily uncomfortable for them. However, both previous results and the studies presented in this paper show that also avoidantly attached people experience loneliness. According to Mikulincer and Shaver, this means “that avoidant people may not deactivate their attachment systems to the point of not caring at all about the absence of supportive relationships” [9] (p. 40). 

As hypothesized, the results of our research indicate that the mechanism explaining the relationships between anxious and avoidant attachment and loneliness is EI. Both insecure attachment styles turned out to be associated with a low level of EI, which in turn translated into increased loneliness. These findings are in line with previous studies, which showed that insecure attachment styles were associated with emotional regulation difficulties [39,40] and that EI prospectively predicted loneliness [25]. Importantly, in the second study, we found this mediation effect when controlling simultaneously for internal representations of self (i.e., self-worth) and others (i.e., benevolence of the people)—a different mechanism behind the attachment–loneliness link, suggested both in theory and in previous research. These results point to EI as a kind of bridge between early childhood and adult interpersonal experiences. Adaptive emotional regulation strategies develop on the foundation of secure attachment with caregivers, and in adult life, they make it possible to form and maintain satisfying social relationships. Avoidant and anxious attachment, by contrast, leads to the development of two radically distinct but, in both cases, maladaptive strategies for managing emotions [41]. In the case of anxiously attached people, it is a hyperactivation strategy, characterized by increased sensitivity to negative emotions from others and excessive focus on one’s own emotional discomfort, which, in practice, makes it difficult to build social relationships with the other person as an equal partner. Avoidantly attached individuals develop a deactivating strategy, aimed at reducing stress through suppression and repression, and when the source of stress is other people—through distancing oneself from them [41]. This is the aftermath of their experiences, which show that others will not be available to soothe their emotions. In practice, both strategies hinder the perception, understanding, and regulation of emotions during social interactions [23], which increase the risk of loneliness [9]. 

In both studies, we also found the hypothesized moderating role of EI in the relationship between anxious attachment and loneliness. It turned out that the strength of the positive relationship between anxious attachment and loneliness decreased with an increase in EI. Furthermore, both studies revealed that at a high level of EI anxious attachment ceases to be significantly related to loneliness. This suggests that uncertainty about one’s own worth and doubts about the possibility of being accepted by others, which are characteristic of anxious attachment, can be effectively reduced through EI resources. This finding is consistent with the existing results of research on emotional regulation in people with high EI. Individuals with high EI show a lower level of negative mood as a result of laboratory-induced stress than those with low EI [42]. Moreover, people high in EI more often use adaptive strategies for coping with stress (e.g., positive reappraisal) than maladaptive emotional regulation strategies (e.g., self-blame) [43]. For example, they interpret stressful situations as a risk rather than a threat and have stronger faith in their ability to cope with such situations [44]. In a social context, in research on adolescents, it was found that EI buffered the impact of traditional victimization on loneliness [28]. In a different study, Nozaki and Koyasu [45] demonstrated that a high level of emotional self-regulation predicted better inhibition of retaliation for ostracism. Given that anxious attachment is associated with low self-esteem [46], our result is also in line with the investigation reported by Kashdan et al. [47], who found that one of the EI features—emotion differentiation—neutralized neural responses (i.e., dACC and anterior insula) to social rejection in people with low self-esteem. These results can be alternatively interpreted as indicating the moderating role of attachment in the relationship between EI and loneliness. Viewed from this perspective, our research shows that the strength of the link between EI and loneliness increases with an increase in the level of anxious attachment. This means that the higher the level of this attachment style, the more effective the reduction of loneliness through EI. This is consistent with the theoretical assumptions concerning anxious attachment. A characteristic feature of anxiously attached people is increased vigilance and sensitivity to signals of social threat combined with higher motivation to obtain support; this can create perfect conditions for using resources associated with recognizing other people’s emotions and regulating one’s own moods for the purpose of restoring social connection. Most importantly, however, both interpretations suggest that EI may effectively protect individuals with chronic attachment-related anxiety from a sense of loneliness. The moderation effect of EI was not present in the case of avoidant attachment—the relationship between this attachment style and loneliness does not change across different levels of EI. This means that in avoidantly attached people, EI does not show regulatory properties enabling the reduction of loneliness. Seeking an explanation for this, it is worth referring to Mikulincer and Shaver’s reflection on loneliness in avoidantly attached people. As these authors observe, while both insecure attachment styles are associated with increased loneliness, avoidant people have low motivation to overcome this state, as a result of which they remain isolated and lonely over time [9]. This is well illustrated by research showing that whereas the level of loneliness in anxiously attached people decreases with age, in avoidantly attached individuals it remains similar over the years [46]. This means that, being weakly motivated to overcome loneliness, avoidant people do not use the resources associated with emotional intelligence for this purpose. Given the characteristics of avoidant attachment (including the low level of trust), it is possible that people with a high level of this style tend to use emotional intelligence to detect potential signs of hostility or other social threats rather than to build close relationships with others. 

### Limitations

While the fact that both studies yielded a consistent pattern of results is an advantage of our research, several limitations should be noted. Firstly, the weaknesses include the cross-sectional design, which does not justify conclusions regarding causality, and the largely retrospective measurement of attachment styles (especially in Study 2), which carries a risk of memory biases. As regards the former limitation, although the proposed interpretation of the relationship between the variables is supported both by the attachment theory and by data on the prospective prediction of loneliness from EI, it is worth performing a longitudinal measurement in the future (optimally, at three time points [48]), particularly in order to replicate the mediation effects detected. Performing this kind of prospective measurement of attachment styles together with EI and loneliness at different developmental stages, starting from childhood, would also allow for reducing the second weakness mentioned above—the risk of memory biases in the evaluation of relations with caregivers. Secondly, we used snowball sampling, which on the one hand allowed us to reach a fairly large group of people and conduct two studies instead of only one, but on the other hand, this sampling resulted in predominantly young and female participants, limiting the possibilities of generalizing the results. In future studies, it is worth reaching more varied samples, testing if the relationships between attachment styles, EI, and loneliness, discussed above, are the same or similar in males, older adults, and people from countries more ethnically diverse than Poland. Finally, both studies included only trait EI; in future research, the mediating and moderating roles of EI abilities should also be investigated (e.g., using performance-based measures of EI).

## 5. Conclusions

Previous research revealed that both anxious and avoidant attachment styles were positively related to loneliness. Little is known, however, about the mediators and moderators of these relationships. The aim of the presented research was to examine if emotional intelligence (EI) could be regarded as a mechanism mediating the relationship between attachment and loneliness. The authors also tested the moderating role of this variable, investigating whether EI was a protective factor against loneliness in insecurely attached individuals. The two studies presented in this article yielded a consistent pattern of results, indicating a dual role of EI in the relationships between attachment styles and loneliness. Firstly, they showed the mediating role of this variable, explaining why insecure attachment styles were linked with a risk of increased loneliness. The results suggest that both anxiously and avoidantly attached individuals are characterized by a low level of ability to regulate their emotions, which, in the interpersonal context, is associated with an increased sense of social isolation. Thus, low EI appears to be an intermediary link in the intensification of loneliness as a result of insecure social experiences in childhood. Furthermore, the present research is the first to find the mediating role of EI while controlling for another mediator of the relationships between attachment styles and loneliness, well documented by now, namely internal representations of self and others. 

Secondly, in the current research, it was found for the first time that EI buffered the effect of anxious attachment on loneliness, which means that the strength of the relationship between anxious attachment and loneliness decreased with an increase in EI. Moreover, when EI is high, anxious attachment—a well-documented predictor of loneliness—is no longer related to this variable. This novel finding has important practical implications. Given that EI can be developed throughout the life span [49], these data suggest that anxiously attached persons can effectively protect themselves from loneliness by training their emotional regulation skills. In other words, highly developed EI can prevent early childhood anxious interpersonal experiences from translating into problems linked with a sense of isolation in adult life. 

## Figures and Tables

**Figure 1 ijerph-19-14831-f001:**
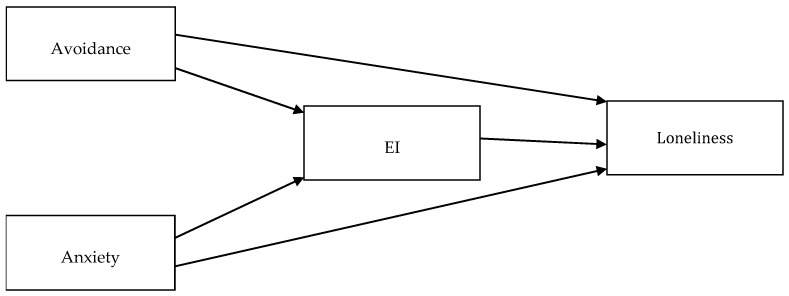
Mediation model with two independent variables and a simple mediator, tested in Study 1.

**Figure 2 ijerph-19-14831-f002:**
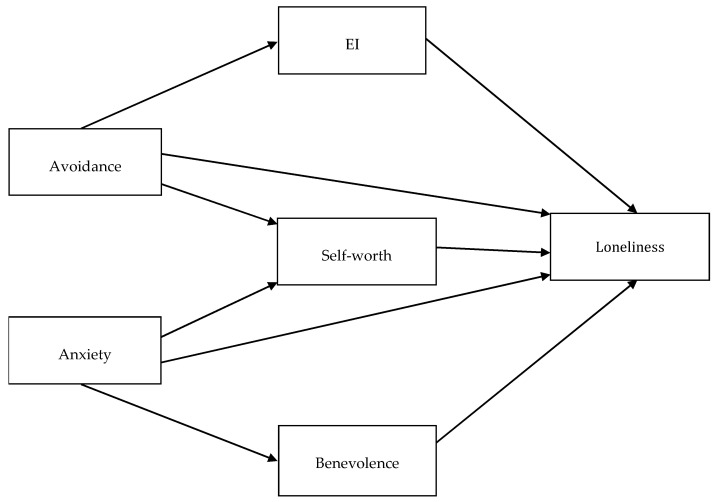
Mediation model with two independent variables and parallel mediators, tested in Study 2.

**Figure 3 ijerph-19-14831-f003:**
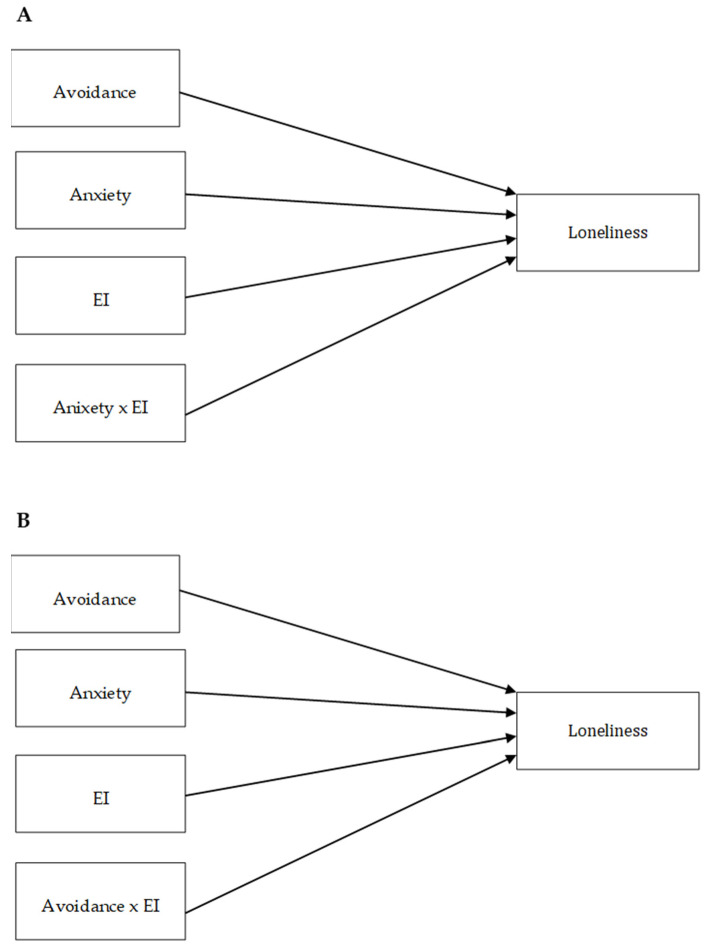
Moderation models for attachment anxiety (**A**) and attachment avoidance (**B**) tested in Studies 1 and 2.

**Figure 4 ijerph-19-14831-f004:**
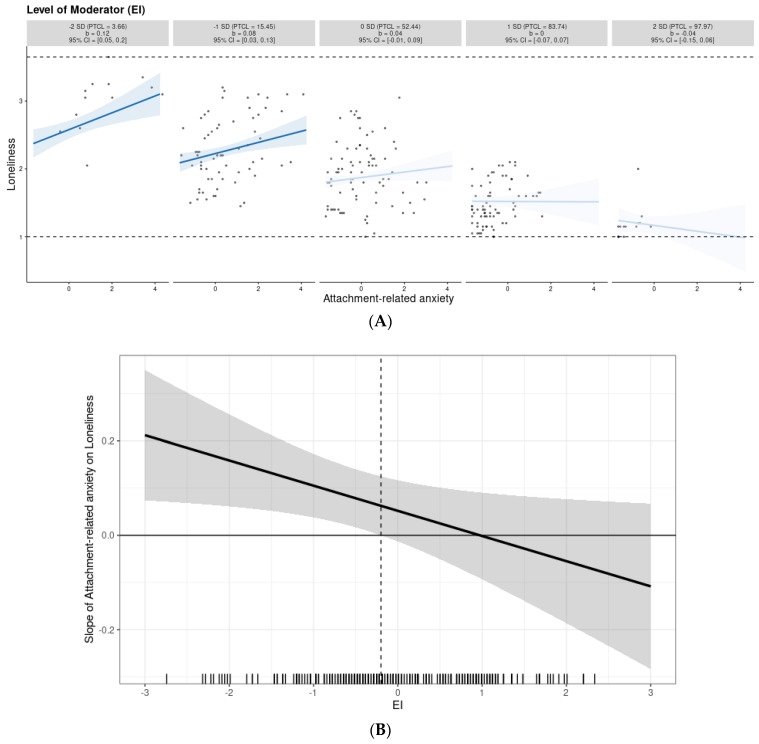
Graphical representation of the moderation analysis in Study 1. The interaction between attachment-related anxiety and EI in predicting loneliness (panel (**A**)) and Johnson–Neyman regions representing the threshold of significance for the effect of attachment-related anxiety on loneliness at different levels of EI (panel (**B**)). Note: The figures were generated using the interActive data visualization tool. The shaded regions in panel B indicate 95% confidence intervals. The effect of attachment-related anxiety on loneliness is statistically significant left of the dashed vertical line (the confidence bands there do not include zero).

**Figure 5 ijerph-19-14831-f005:**
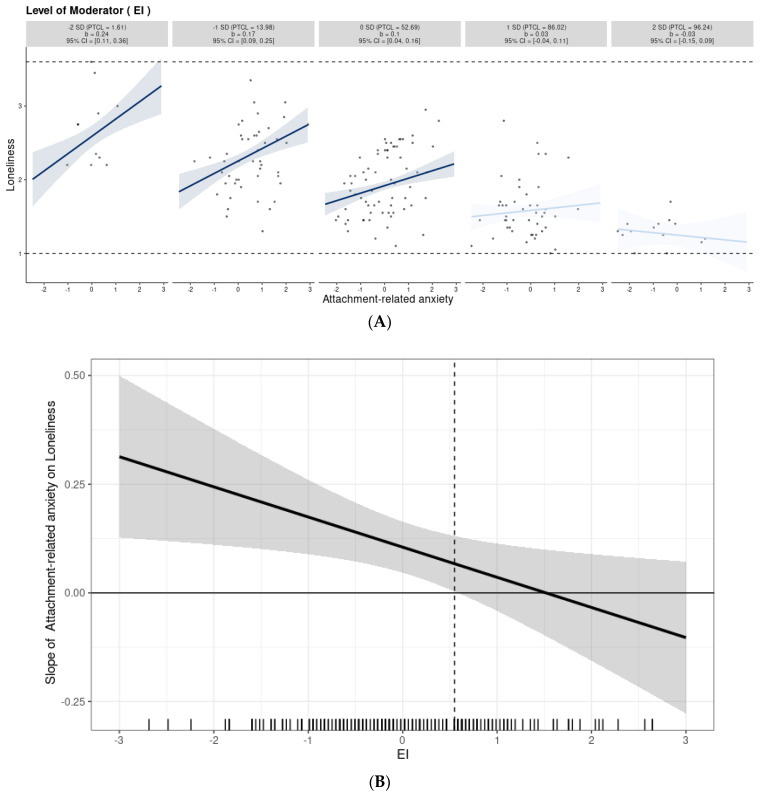
Graphical representation of the moderation analysis in Study 2. The interaction between attachment-related anxiety and EI in predicting loneliness (panel (**A**)) and Johnson–Neyman regions representing the threshold of significance for the effect of attachment-related anxiety on loneliness at different levels of EI (panel (**B**)). Note. The figures were generated using the interActive data visualization tool. The shaded regions in panel B indicate 95% confidence intervals. The effect of attachment-related anxiety on loneliness is statistically significant left of the dashed vertical line (the confidence bands there do not include zero).

**Table 1 ijerph-19-14831-t001:** Participants’ sociodemographic characteristics.

Variable		Study 1 *N* = 246		Study 2 *N* = 186	
		*n*	%	*n*	%
Gender	Female	215	87.4	152	81.7
	Male	29	11.8	34	18.3
	Non-binary	2	0.8	-	-
Place of residence	Village	52	21.1	51	27.4
	City with less than 100,000 inhabitants	99	40.2	40	21.5
	City with more than 100,000 inhabitants	95	38.6	95	51.1
Relationship status	Single	82	33.3	78	41.9
	In a relationship	164	66.7	108	58.1
Education	Below secondary	-	-	9	4.8
	Secondary	-	-	102	54.8
	Higher	-	-	75	40.3
Employment status	Unemployed	19	7.7		-
	Retired	2	0.8		-
	Student	140	56.9		-
	Employed	85	34.6		-

**Table 2 ijerph-19-14831-t002:** Descriptive statistics.

Variable	Valid	*M*	*SD*	Skewness	Kurtosis	Min.	Max.	Cronbach’s α
Study 1								
Avoidance	246	2.72	0.99	0.57	0.87	1.00	6.96	0.89
Anxiety	246	2.66	1.28	0.95	0.59	1.00	7.00	0.86
EI	246	4.56	1.02	−0.07	−0.33	1.77	6.93	0.92
Loneliness	246	1.90	0.60	0.59	−0.40	1.00	3.65	0.90
Study 2								
Avoidance	186	3.74	1.29	0.33	−0.46	1.10	6.80	0.93
Anxiety	186	3.48	1.04	−0.03	−0.22	1.00	6.40	0.87
EI	186	4.55	0.83	0.16	0.10	2.33	6.73	0.89
Self-worth	186	4.00	1.26	−0.34	−0.66	1.00	6.00	0.85
Benevolence	186	3.86	0.89	−0.50	−0.02	1.00	5.75	0.69
Loneliness	186	1.94	0.55	0.44	−0.43	1.00	3.60	0.91

**Table 3 ijerph-19-14831-t003:** Zero-order correlations between the variables (Study 1).

		Avoidance	Anxiety	EI	Loneliness
Avoidance	Pearson’s *r*	-			
	*p* value	-			
Anxiety	Pearson’s *r*	0.55	-		
	*p* value	<0.001	-		
EI	Pearson’s *r*	−0.53	−0.51	-	
	*p* value	<0.001	<0 .001	-	
Loneliness	Pearson’s *r*	0.54	0.51	−0.73	-
	*p* value	<0 .001	<0.001	<0 .001	-

**Table 4 ijerph-19-14831-t004:** Zero-order correlations between the variables (Study 2).

		Avoidance	Anxiety	EI	Loneliness	Self-worth	Benevolence
Avoidance	Pearson’s *r*	-					
	*p* value	-					
Anxiety	Pearson’s *r*	0.24	-				
	*p* value	0.001	-				
EI	Pearson’s *r*	−0.56	−0.31	-			
	*p* value	<0.001	<0.001	-			
Loneliness	Pearson’s *r*	0.44	0.38	−0.69	-		
	*p* value	<0.001	<0.001	<0.001	-		
Self-worth	Pearson’s *r*	−0.34	−0.31	0.76	−0.65	-	
	*p* value	<0.001	<0.001	<0.001	<0.001	-	
Benevolence	Pearson’s *r*	−0.205	−0.04	0.28	−0.24	0.20	-
	*p* value	0.005	0.576	<0.001	0.001	0.007	-

**Table 5 ijerph-19-14831-t005:** Mediation analysis for Study 1.

Type	Effect	β	* SE *	95% CI		*p*
				LL	UL	
Indirect	Avoidance ⇒ EI ⇒ Loneliness	0.21	0.03	0.08	0.18	<0.001
	Anxiety ⇒ EI ⇒ Loneliness	0.18	0.02	0.05	0.12	<0.001
Component	Avoidance ⇒ EI	−0.37	0.08	−0.52	−0.22	<0.001
	EI ⇒ Loneliness	−0.58	0.03	−0.40	−0.28	<0.001
	Anxiety ⇒ EI	−0.30	0.05	−0.34	−0.15	<0.001
Direct	Avoidance ⇒ Loneliness	0.16	0.03	0.04	0.16	0.002
	Anxiety ⇒ Loneliness	0.12	0.03	0.01	0.11	0.026
Total	Avoidance ⇒ Loneliness	0.37	0.04	0.15	0.30	<0.001
	Anxiety ⇒ Loneliness	0.30	0.03	0.08	0.20	<0.001

**Table 6 ijerph-19-14831-t006:** Mediation analysis for Study 2.

Type	Effect	β	* SE *	95% CI		* p *
				LL	UL	
Indirect	Avoidance ⇒ EI ⇒ Loneliness	0.19	0.02	0.04	0.12	<0.001
	Avoidance ⇒ Self-worth ⇒ Loneliness	0.08	0.01	0.01	0.06	0.018
	Avoidance ⇒ Benevolence ⇒ Loneliness	0.01	0.01	−0.01	0.02	0.397
	Anxiety ⇒ EI ⇒ Loneliness	0.07	0.01	0.01	0.07	0.016
	Anxiety ⇒ Self-worth ⇒ Loneliness	0.07	0.01	0.01	0.07	0.014
	Anxiety ⇒ Benevolence ⇒ Loneliness	−0.00	0.003	−0.01	0.006	0.944
Component	Avoidance ⇒ EI	−0.51	0.04	−0.41	−0.24	<0.001
	EI ⇒ Loneliness	−0.38	0.06	−0.36	−0.12	<0.001
	Avoidance ⇒ Self-worth	−0.28	0.07	−0.41	−0.13	<0.001
	Self-worth ⇒ Loneliness	−0.30	0.04	−0.19	−0.05	<0.001
	Avoidance ⇒ Benevolence	−0.21	0.05	−0.24	−0.04	0.005
	Benevolence ⇒ Loneliness	−0.06	0.03	−0.10	0.04	0.332
	Anxiety ⇒ EI	−0.18	0.05	−0.24	−0.05	0.002
	Anxiety ⇒ Self-worth	−0.24	0.09	−0.45	−0.12	<0.001
	Anxiety ⇒ Benevolence	0.01	0.07	−0.12	0.14	0.919
Direct	Avoidance ⇒ Loneliness	0.10	0.03	−0.02	0.10	0.176
	Anxiety ⇒ Loneliness	0.17	0.03	0.03	0.14	0.003
Total	Avoidance ⇒ Loneliness	0.37	0.03	0.10	0.21	<0.001
	Anxiety ⇒ Loneliness	0.29	0.03	0.09	0.22	<0.001

**Table 7 ijerph-19-14831-t007:** Moderation analysis for Study 1.

Predictor	β	*SE*	*t*	95% CI		*p*
				LL	UL	
Intercept		0.09	17.24	1.41	1.60	<0.001
Avoidance	0.17	0.03	3.16	0.04	0.16	0.002
Anxiety	0.09	0.03	1.57	0.01	0.09	0.118
EI	−0.59	0.03	−11.41	−0.41	−0.29	<0001
Anxiety × EI	−0.10	0.02	−2.20	−0.08	−0.004	0.029
Model summary			*R*^2^ = 0.58 *F* (4, 241) = 84.41, *p* < 0.001			
Intercept		0.07	17.24	1.59	1.87	<0.001
Avoidance	0.15	0.03	2.80	0.03	0.15	0.005
Anxiety	0.12	0.02	2.32	0.008	0.10	0.021
EI	−0.59	0.03	−11.33	−0.41	−0.29	<0.001
Avoidance × EI	−0.07	0.02	−1.62	−0.07	0.01	0.107
Model summary			*R*^2^ = 0.58 *F* (4, 241) = 83.10, *p* < 0.001			

**Table 8 ijerph-19-14831-t008:** Moderation analyses for Study 2.

Predictor	β	*SE*	*t*	95% CI		*p*
				LL	UL	
Intercept		0.10	17.16	1.58	2.00	<0.001
Avoidance	0.08	0.03	1.29	−0.02	0.09	0.199
Anxiety	0.19	0.03	3.53	0.04	0.16	0.001
EI	−0.61	0.04	−9.66	−0.49	−0.32	<0.001
Anxiety × EI	−0.13	0.03	−2.41	−0.15	−0.01	0.017
Model summary			*R*^2^ = 0.52 *F* (4, 181) = 49.82, *p* < 0.001			
Intercept		0.11	14.15	1.35	1.78	<0.001
Avoidance	0.05	0.03	0.78	−0.03	0.07	0.437
Anxiety	0.19	0.03	3.45	0.04	0.16	0.001
EI	−0.60	0.04	−9.44	−0.48	−0.32	<0.001
Avoidance × EI	−0.07	0.03	−1.22	−0.08	0.02	0.226
Model summary			*R*^2^ = 0.51 *F* (4, 241) = 47.62, *p* < 0.001			

## Data Availability

The anonymized data are available from the corresponding author upon reasonable request.

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
