# Peer review of "Attached but Lonely: Emotional Intelligence as a Mediator and Moderator between Attachment Styles and Loneliness"

_ijerph, 2022, doi:10.3390/ijerph192214831_

Round 1

Reviewer 1 Report

Just a few comments as the paper is sound and informative. You estimate 2 models using 2 samples to analyze the relation between Loneliness with Avoidance, and Anxiety, using a third variable, Emotional Intelligence that has a mediator and a moderator influence in the original relation.

I think you should be critic with the samples obtained, as they are mostly composed of females and students, and with an age interval that really is much less than the proposed, as they are mostly young. Also, any Randomness priciple is absent in the experimental design.

I would suggest that you explain the methodology used. Mainly it is a Path Analysis model, with 2 endogenous variables, and using an interaction fo study the moderation effect of EI. A few lines in this part would be welcome by the readers (and a graph diagram of the model).

As you add/average the original ordinal variables used to obtain the variables in the model, you treat them as numerical. A hint about this would also be welcome.

Also, have you try to use both samples in an unique model? (using the common variables).

You hardly do not describe the samples for the rest of the variables collected. For example, the Gender could have an influence on the relations estimated?

Please consider these questions as curiosity and not of criticism of the manuscript.

Reviewer 2 Report

The article reports on research into the mediating role of emotional intelligence between attachment and loneliness.

The introduction provides sufficient background on the topic and previews major points. Both research design and analysis are adequate

The authors are encouraged to justify why they chose snowball sampling as the most suitable for the study.

In the conclusions section, the authors stress the consistency of the results with previous research. However, the motivation for this research or why it is novel in terms of previous studies is not made sufficiently clear.
